# Investigating the Effect of Supercritical Carbon Dioxide Treatment on the Rheological, Thermal, and Functional Properties of Plum (*Prunus domestica* L.) Kernel Protein Isolates

**DOI:** 10.3390/foods12040815

**Published:** 2023-02-14

**Authors:** Mohd Aaqib Sheikh, Charanjiv Singh Saini, Harish Kumar Sharma

**Affiliations:** 1Department of Food Engineering and Technology, Sant Longowal Institute of Engineering and Technology, Longowal 148106, India; 2Department of Chemical Engineering, National Institute of Technology, Agartala 799046, India

**Keywords:** plum kernel protein isolate, supercritical carbon dioxide, rheology, microstructure, functional properties, waste utilization

## Abstract

Plum kernels are a promising source of dietary proteins that are irretrievably lost during processing. The recovery of these underexploited proteins could be eminently vital for human nutrition. Plum kernel protein isolate (PKPI) was prepared and exposed to a targeted supercritical carbon dioxide (SC-CO_2_) treatment to diversify its effectiveness in industrial applications. The impacts of SC-CO_2_ treatment at different processing temperatures (30–70 °C) on dynamic rheology, microstructure, thermal, and techno-functional characteristics of PKPI were investigated. The results revealed that the dynamic viscoelastic characteristics of SC-CO_2_-treated PKPIs showed higher storage modulus, loss modulus, and lower tan δ value than native PKPI, indicating greater strength and elasticity of the gels. Microstructural analysis showed that the proteins experienced denaturation at elevated temperatures and resulted in the formation of soluble aggregates, which increased the heat requirement for thermal denaturation of SC-CO_2_-treated samples. SC-CO_2_-treated PKPIs demonstrated a decline of 20.74% and 30.5% in crystallite size and crystallinity. PKPIs treated at 60 °C showed the highest dispersibility, which was 1.15-fold higher than the native PKPI sample. SC-CO_2_ treatment offers a novel path to improve the techno-functional properties of PKPIs and extend its use in food and non-food applications.

## 1. Introduction

Proteins are extensively used as food ingredients for their nutritional richness and functional characteristics that play a crucial role in various food formulations [1]. Generally, they are utilized in various food applications because of their desirable functional properties, including emulsifying and foaming agents, texture and color improvers, gelling and whipping capabilities, and oil/water absorption capabilities that affect consumer perception of food items [2]. Protein deficiency is becoming a severe global issue that affects both developing and developed nations, primarily because of population increase and the rising demand for aquaculture, feed, and industrial materials [3]. Nutritionists are exploring plant protein substitutes for people and feedstock since they are more easily accessible, digestible, affordable, and pose fewer health risks than animal protein sources [4]. Due to the excellent nutritional profile, low cost, and increasing consciousness about environmental safeguards, animal welfare, and personal health, the demand for plant proteins in food and non-food products has progressively increased globally over the past few decades [5]. In response to the rising demand, efforts to extract proteins from agro-industrial wastes are seen as profitable, sustainable, and advantageous alternatives for improving the supply of essential nutrients [6].

Plums (*Prunus domestica* L.), a multipurpose crop in the *Rosacea* family and genus *Prunus*, are amongst the most significant fruits in terms of customer preference, delicious taste, and nutritional properties [7], with global production in 2020 reaching 12.1 million tons [8]. In addition to being consumed fresh, a significant portion is used to develop value-added products [9], which results in the generation of a huge quantity of plum kernels that constitute more than 60% of the fruit weight and are primarily wasted and undervalued [10,11]. Plum kernels are highly nutritious and a cost-effective source of high-quality nutrients that are irretrievably wasted during processing [12]. These underutilized kernels contain an abundance of proteins (35.24%) and free peptides [13], offering a novel and potentially inexpensive source of material for the food, cosmetics, and pharmaceutical industries [14]. The amino acid composition of plum kernel proteins revealed a significant proportion of hydrophobic and aromatic amino acids, and thirteen peptides with antioxidant and angiotensin-converting enzyme inhibitor properties were identified [15]. The recovery of these proteins could be a non-conventional source with typical features to scavenge hydroxyl radicals and bioactive peptides beyond their nutritional and functional properties [16]. The isolation of plum kernel proteins could be a promising source of high-quality proteins with a plethora of desirable characteristics in the industries for product formulations [17]. When a protein is used as a constituent in the food system, its functionality might be more significant than its nutritional benefits because functional characteristics are crucial for the processing and formulation of food products [18]. The inclusion of plant proteins as a food ingredient has been restricted due to limitations in functional characteristics, and it is essential to assess the quality and functional attributes of plant proteins. The functional characteristics are distinct features closely linked to molecular composition and protein structures [5]. To increase the utilization, the structure of these proteins needs to be changed for the suitability of these proteins as a functional components in different food products [19]. Studies on modifying the functional properties of proteins using physical, chemical, and biological approaches have been extensively reported in recent years. Modifying protein structures adds novel functionalities to the unaltered protein structure [20].

In recent years, supercritical carbon dioxide (SC-CO_2_) technology has gained recognition as an eco-friendly approach and seems quite promising in modifying the protein structure [21], owing to its practical applications in biopolymer processing for commercial and scientific purposes [22]. Due to its favorable characteristics, such as non-toxicity, inertness, affordability, non-polluting, non-flammability, liquid-like density, and gas-like diffusivity, SC-CO_2_ has gained prominence as a substitute for traditional techniques [23,24]. At the industrial level, SC-CO_2_ has proven to be a very effective way of extracting, purifying, and changing the structural characteristics of biomolecules [25]. The zero surface tension of SC-CO_2_ induces complete and quick wetting, allowing the penetration of complex structures followed by structural alterations due to the depletion or redistribution of nonpolar compounds [26]. The application of supercritical technology offers noticeably high efficiency for separating α-Lactalbumin and *β*-Lactoglobulin and leads to the development of novel and appealing products with bioactive qualities [27]. SC-CO_2_ technologies have become a substitute for traditional processing techniques that have earned keen interest for their potential use in modifying the morphological, structural, and techno-functional properties of biopolymers [24]. Zhong and Jin [28] found that SC-CO_2_ treatment caused structural changes in milk proteins and improved the rheological behavior including gelling properties of whey proteins in water. The significant improvement in gel strength of the SC-CO_2_-treated whey proteins could be ascribed to dissolved supercritical carbon dioxide that interacted with proteins, altered the interfacial properties and caused irreversible structural changes after carbon dioxide depletion [28]. The use of SC-CO_2_ technology appears to be a novel strategy that could be an interesting alternative for reducing the particle diameter, lowering the pseudoplastic nature, and preserving the nutritional quality of the food products [29]. According to Maheshwari et al. [30], using the SC-CO_2_ extraction technology can significantly decrease the off-flavors from soy proteins without affecting the nutritional profile. SC-CO_2_ treatment improved the protein functionalities such as emulsifying and rheological properties by causing a structural and conformational modification of whey proteins [31]. SC-CO_2_ is a promising approach to enhance the functional characteristics such as foaming and rheological properties of egg white protein, by making the protein structure more flexible and looser [26]. The structural changes brought about by SC-CO_2_ treatment increased the applicability of pea protein isolates in food industries owing to better techno-functional properties than native untreated samples [31]. Nonetheless, no research has investigated the effect of SC-CO_2_ treatment on the rheological and thermal properties of plum kernel protein isolates. Therefore, this work is aimed to examine the effect of SC-CO_2_ treatment under various processing conditions on the rheological, thermal, and functional properties of plum kernel protein isolates (PKPI). It is anticipated that the scientific knowledge generated through this study will make it possible to employ PKPIs in the food industries in a specific target and systematic way.

## 2. Materials and Methods

### 2.1. Materials

Plum kernels were purchased from the Central Institute of Temperate Horticulture (ICAR), Srinagar, India. The kernels were kept in the freezer till further investigation. All chemicals employed for the study were bought from Sigma Aldrich Chemicals, Chandigarh, India. Millipore (Elix, Merck, Progard TS2, PR0G0T0S2, Vikhroli (E), Mumbai, India) purified water was used for analysis.

### 2.2. Methods

#### 2.2.1. Preparation of Plum Kernel Protein Isolate (PKPI)

Plum kernel meal was prepared by adopting the method reported by Sheikh and Saini [6]. Plum kernel protein isolate was prepared from defatted plum kernel meal by following the standardized procedure of Sheikh et al. [32] and Xue et al. [33]. The precipitated protein was neutralized with 0.1 N NaOH, freeze-dried, packed in air-tight plastic pouches, and stored in a refrigerator till further investigation. The meal was added to the alkaline water (pH 10) at a 1:10 (*w*/*w*) ratio and stirred for 15 min. The slurry was kept in a shaking water bath at 40 °C for 2 h. After centrifuging the mixture at 8000× *g* for 15 min, the pH of the collected supernatant was brought down to 4.2 by adding 1 N HCl and left for 2 h. The precipitated proteins were separated, neutralized with 0.1 N NaOH, freeze-dried, packed in air-tight plastic pouches, and stored in a refrigerator until further investigation.

#### 2.2.2. Supercritical Carbon Dioxide Treatment

SC-CO_2_ treatment apparatus (Waters-SFE-500 System, Milford, MA, USA) used for the treatment of PKPI was based on the system demonstrated in our previous study [32]. An amount of 25 g of PKPI powder was contained in a sample basket and placed inside the treatment kettle with the system temperature ranging from 30 to 70 °C, carbon dioxide at a mass flow rate of 20 g/min, a pressure of 20 MPa, and processing time of 60 min. The varying values of treatment temperatures of 30, 40, 50, 60, and 70 °C were based on the previous trials. Treated PKPI samples removed from the SC-CO_2_ treatment kettle were kept at ambient conditions for 4 h to release the residual carbon dioxide and stored in the refrigerator till further analysis. The untreated PKPI served as a control sample.

#### 2.2.3. Rheological Properties

A Paar Physica MCR 300 Rheometer (Gaz, Austria) equipped with a parallel plate measuring geometry was employed for dynamic oscillatory measurements of PKPI samples by following the procedure outlined by Malik and Saini [19]. The plates had a diameter of 30 mm and were positioned with a gap of 1 mm. The plate margins were covered with low-viscosity silicone to prevent sample dehydration. A Haake (DC5B3) water thermo-circulator with a Peltier temperature control unit was used to regulate the temperature. Amplitude and frequency measurements were conducted to determine the linear viscoelastic region (LVR). The protein solution (20%, *w*/*v*) was heated at a rate of 10 °C/min from 25 °C to 95 °C, kept there for 15 min, and then cooled down slightly to 25 °C at the same rate. The storage modulus (G’), loss modulus (G”), and loss tangent (tan δ) values were computed as a function of temperature and time. The rheological behavior was examined by correlating the dependence of G’ and G” with frequency after the sample cooling. Storage and loss modulus were measured as a function of frequency, while a frequency sweep was carried out at 25 °C in the 0.05–10 Hz range with a constant strain of 0.1%.

#### 2.2.4. Thermo-Gravimetric Analysis (TGA)

A Perkin Elmer TGA (TGA-4000, Shelton, CT, USA) was employed for the thermal gravimetric analysis of PKPI samples by following the procedure described by Mir et al. [5]. The thermograms were generated in the range of 30–480 °C at a rate of 10 °C/min, and nitrogen was used as a cooling gas at 20 mL/min.

#### 2.2.5. X-ray Diffraction (XRD) Pattern

To determine the crystalline state of untreated and SC-CO_2_-treated PKPI samples, XRD patterns were taken using an X-ray diffractometer (Model: D8 Advance DAVINCI, Bruker AXS Inc., Madison, WI, USA) by following the method outlined by Malik and Saini [34]. Diffractograms were recorded between 5–40° (2θ) at a rate of 3.2°/min with a step size of 0.0131°. The collected data were processed using XPert High Score plus software to determine the crystallinity.

#### 2.2.6. Microstructural Analysis

The morphology of the untreated and SC-CO_2_-treated PKPI samples was observed by field emission scanning electron microscopy (JEOL JSM 7610 F PLUS, Musashino, Tokyo, Japan) based on the method of Mir et al. [5]. The freeze-dried PKPI samples were coated with gold particles using a sputter coater before being examined.

#### 2.2.7. Functional Characterization of PKPI

##### Dispersibility

The dispersibility of PKPI was assessed by following the approach of Mir et al. [35] with minor modifications. A 50 mL measuring cylinder containing 3 g of protein isolate was filled with 30 mL of distilled water. The mixture was incubated for 2 h after being agitated (Tarsons Digital Spinot, MC-02, DN19012500, Mumbai, India) at 600 rpm for 30 min. The volume of the settled particles was measured, and dispersibility was measured as follows:$$Dispersibility\;\left(\%\right)=\frac{Total\;volume-settled\;volume}{Total\;volume}\times 100\;$$

##### Least Gelation Concentration (LGC)

LGC of PKPI was determined using the method of Ohizua et al. [36] with slight modification. The protein dispersions of 6, 8, 10, 12, 14, and 16% prepared in 5 mL distilled water was heated at 90 °C for 1 h in a shaking water bath (Narang Scientific Works Pvt. Ltd., New Delhi, India). The resulting coagulum was cooled at 0 °C in an ice bath and stood for 2 h at 4 °C. The lowest protein concentration or gelation endpoint was determined as the concentration at which the coagulum did not fall or slip when the tube was inverted.

#### 2.2.8. Water Activity

Around 1.5 g of the untreated and SC-CO_2_-treated PKPI samples were placed in the measuring cell of the water activity meter (Rotronic, Hygrolab 2, Bassersdorf, Schweiz, Switzerland), and water activity (a_w_) was determined at ambient temperature.

#### 2.2.9. Data Analysis

Each experiment was carried out at least three times, and the data depicted in the figures and tables represent the mean ± standard deviation. A statistical application IBM, SPSS Statistics (Version 26) was employed for the data analysis. Duncan’s multiple range test (DMRT) evaluated significant differences among the mean values with significance defined as *p* < 0.05 at 95% confidence level.

## 3. Result and Discussion

### 3.1. Effect of SC-CO_2_ Treatment on Dynamic Rheology of PKPIs

The rheological characteristics of proteins during temperature change are beneficial in controlling texture, and mouth feels in various food applications. The viscoelastic characteristics of native and SC-CO_2_-treated PKPIs at various processing temperatures were investigated by observing the change in storage modulus (G’), loss modulus (G”), and loss tangent (tan δ) during three phases: the first phase, the heating step, which took place from 25 to 95 °C, followed by a second phase, a 15 min hold at 95 °C and the third phase of cooling phase from 95 to 25 °C. G’ and G” are rheological characteristics that demonstrate gel strength. Figure 1(i) illustrates the changes in dynamic rheological characteristics of native and SC-CO_2_-treated PKPIs at various processing temperatures. The G’ of all SC-CO_2_-treated PKPIs were greater than those of their native PKPI, and SC-CO_2_-treated PKPIs at 70 °C (ST 70) showed the highest G’ value indicating gels with greater strength and elasticity of the structure contributing to the three-dimensional network due to protein aggregations. G’ was low and nearly constant until a temperature of around 70 °C was reached. The protein molecules at this point were denatured and exposed to the hydrophobic residues as a preliminary step for gel formation. The temperature at which G’ enhances and becomes higher than the background noise is called gelation temperature (T_Gel_) and signifies a change from a liquid-like state to a solid-like state [37]. T_Gel_ is typically higher than the denaturation temperature, and denaturation is a prerequisite for heat-induced gelation of globular proteins [38]. All SC-CO_2_-treated PKPI samples showed a higher T_Gel_ value when compared to the native sample. The PKPI samples processed at 70 °C (ST 70) displayed the highest T_Gel_ at around 74 °C (Figure 1A). After the heating phase, the successive cooling phase causes further improvement in the storage modulus of the native and SC-CO_2_-treated PKPI samples, which is called gel reinforcement (G_Rf_) and indicates the formation of cross-linking and rearrangements of the network structures [39]. G_Rf_ is a characteristic of protein gels and is typically linked to the realignment of attractive forces within the gel network, such as Van der Waals and hydrogen bonding between proteins [37]. The G” or loss modulus indicates the viscous component and reveals interactions that do not contribute to the three-dimensional network. As illustrated in Figure 1B, loss modulus values increased steadily across all three stages. The larger G’ and G” levels found in SC-CO_2_-treated PKPIs at elevated processing temperatures could be the result of uncovering the significant number of hidden hydrophobic and hydrogen bonds, which then participate in inter-molecular interactions.

The results suggested that all PKPIs followed a three-stage gelation process comprising: (1) protein denaturation leading to the exposure of hydrophobic residues; (2) intermolecular hydrophobic interaction of the unfolded proteins (aggregation); and (3) agglomeration of aggregates into a network structure. The findings are in line with Sun and Arntfield [39], who claimed that the development of hydrophobic interactions and hydrogen bonds were responsible for the formation and stability of pea protein isolate gels. It was assumed that carbon dioxide in the supercritical state penetrated the interior of the protein structure, disrupted non-covalent bonds, exposed the hydrophobic moieties previously buried within the interior of the protein structure, and improved cross-linking between protein molecules. Sheng et al. [40] reported that under a supercritical state, carbon dioxide broke some of the protein structure’s disulfide bridges and exposed the buried sulfhydryl groups and other hydrophobic moieties. Hence, it is plausible that the same interactions influenced the development and persistence of SC-CO_2_-treated PKPI gels.

The impact of SC-CO_2_ treatment on gel structure may also be evidenced by the shift in phase angle or tan δ. Phase angle or loss tangent is a key marker for identifying a gel formation and is employed to evaluate the contribution of viscous and elastic constituents in the resulting gel structure [41]. Consequently, a variance in tan δ ought to reflect the kind of network created, with lower tan δ values indicating superior three-dimensional networks [19]. As shown in Figure 1C, tan δ started to decline progressively during the heating and cooling phases, indicating a stable gel had developed from the start of the chilling phase, and changes during the cooling phase strengthened the elastic and viscous components of networks. All SC-CO_2_-treated PKPI samples showed a lower tan δ value when compared to the native sample. The lowest tan δ value was found in PKPI samples treated at 60 °C (ST 60), indicating a greater capacity for forming protein–protein interaction than native PKPI samples because lower tan δ values for gels indicate elastic structures due to the stronger interactions between protein molecules, including hydrophobic interactions and disulfide bonds [39]. The improvement in elastic structures might be attributed to the unfolding of protein structures during the SC-CO_2_ treatment that exposed buried negatively charged sites on the surface and strengthened the interparticle electrostatic repulsions [40]. Similarly, Zhong and Jin [26] found that SC-CO_2_ treatment caused structural changes in milk proteins and improved the rheological behavior including gelling properties of whey proteins in water. The significant improvement in gel strength of the SC-CO_2_-treated proteins could be ascribed to dissolved SC-CO_2_ that interacted with proteins, altered the interfacial properties, and caused irreversible structure changes after carbon dioxide depletion [26]. When the treatment was shifted to an elevated temperature of 70 °C (ST 70), an improvement in tan δ was noticed, indicating inferior gel-forming abilities of the PKPI samples with more viscous components [37]. This might be attributed to the reduced hydrophobicity of PKPI samples due to protein aggregation, which restricts linkages between proteins because the carbon dioxide under a supercritical state at a higher temperature combines with water and produces carbonic acid, which decreases the negative charge of proteins by increasing hydrogen ions and enhances aggregation [31].

A frequency sweep was conducted to study more about the gel structure (Figure 1D,E). As a function of frequency, the storage modulus (G’) and loss modulus (G”) are effective evaluation approaches for exploring protein interactions and calculating the elastic and viscous components in gel systems [40]. G′ measures the compactness or the number of cross-links inside gel structures and indicates the energy stored in the material. In contrast, G” indicates the energy loss of dynamic oscillations [42]. Both G’ and G” of native and SC-CO_2_-treated PKPIs increased with increasing frequency, demonstrating the dependency of gel on frequency and characteristics of a weak gel system. However, the magnitudes of G’ were noticeably greater than those of G”, suggesting an elastic behavior of gel, owing to the stronger inter-molecular networks and protein–protein interactions amongst the networks [40]. Compared to the native PKPI sample at any frequency, all SC-CO_2_-treated PKPI samples showed a lower G’ and G” moduli, implying that the SC-CO_2_ treatment disrupted the protein–protein interactions, destabilized the entanglement network to promote protein molecule slippage. In the case of SC-CO_2_-treated PKPIs, the dependence of gels on the frequency reduced as the treatment temperature increased, suggesting that increasing temperature resulted in stronger gels. The results follow the findings of Sheng et al. [40], who concluded that the high-pressure carbon dioxide treatment on liquid whole eggs caused interference between protein interactions and disruption of the entanglement network. Similarly, Ding et al. [26] revealed that supercritical carbon dioxide treatment altered the viscoelastic nature of egg white protein due to the unfolding of hydrophobic groups under the synergistic effects of high pressure and carbon dioxide, which enhanced the cross-linking between protein molecules.

### 3.2. Thermo-Gravimetric Analysis (TGA)

TGA records various phases of mass losses resulting from chemical changes occurring during the protein molecule’s thermal decomposition. In contrast, characterizing proteins employing TGA, the increase in temperature affects protein molecules in several ways, including the evaporation of water, crystallization, sublimation, fusion, breakage of disulfide bonds, the liberation of low molecular weight volatile compounds, dehydration, and the degradation of the material [34]. TGA was used to evaluate the thermal stability of PKPIs against temperature (Figure 2). Both the native and SC-CO_2_-treated PKPIs showed weight loss in different stages, indicating that the loss in mass is time–temperature dependent. The first stage step, around 100 °C, was associated with minimal weight loss and was ascribed to the evaporation of the water and volatilization of lower molecular weight compounds, accompanied by the structural changes due to the high temperatures [19]. The second stage noticed in a temperature region of 220–400 °C was associated with maximum weight loss and was ascribed to the breakage of molecular interactions, including disulfide bridges, covalent and non-covalent bonds, followed by the complete degradation of protein structure accompanied by the release of gases [41]. Generally, the second step is characterized as a thermal degradation step, during which the bound water inside the proteins begins to lose, and the hydroxyl groups are also destroyed, resulting in the maximum weight loss of the PKPIs.

The further weight loss due to the increasing temperature resulted in the oxidation of the PKPI sample under airflow. Compared to the native PKPI, all SC-CO_2_-treated PKPIs were slightly more thermally resistant, indicating an increased heat requirement for thermal denaturation of SC-CO_2_-treated PKPIs. The higher thermal stability in PKPIs could be attributed to the structural modification and subsequent cross-linking of denatured protein molecules due to SC-CO_2_ treatment at a higher temperature reducing the tendency to lose weight. Overall, under the identical programmed circumstances of temperature increase, the SC-CO_2_-treated PKPIs at 70 °C were more thermally stable than other PKPI samples, which could be attributed to the greater solubility of carbon dioxide at higher temperatures that enhance protein aggregation by reducing the electrostatic repulsion among the adjacent molecules and allowing them to combine via hydrophobic and non-covalent bonding and adds to the thermal stability of the proteins. Similarly, Mudasir et al. [43] attributed the decrease in weight loss during thermal processing to the aggregation of protein after the sonication at 30 min. Carbon dioxide in the supercritical state penetrated the interior of the protein structure and disrupted non-covalent bonds, which resulted in a considerable change in protein structures after the SC-CO_2_ treatment [44]. Shen et al. [40] noticed that the formation of soluble protein aggregates during thermal treatments led to a slight increase in the thermal stability of proteins. The findings were also consistent with Xu et al. [26] and Zhang et al. [45], who reported that heat treatment enhanced the protein’s thermal stability by modifying the structural characteristics due to enhanced protein–protein interactions, followed by protein aggregation and leading to the formation of an extra bunched structure with superior thermal stableness. Murtaza et al. [24] suggested that the carbon dioxide under a supercritical state reacts with the amide groups of some amino acid residues to form amino bonds, resulting in considerable changes in protein conformation. Overall, the findings demonstrated that SC-CO_2_-treated PKPIs were more thermally resistant than native PKPIs, emphasizing their applicability to foods that are subjected to heat processing.

### 3.3. Effect of SC-CO_2_ Treatment on the Microstructure of PKPI

Scanning electron microscopy (SEM) is an effective surface scanning tool with strong magnification capabilities that could assist in gathering precise information on surface morphology, structure, and modifications of protein characteristics [2]. SEM examined the microstructures of the native and SC-CO_2_-treated PKPI samples for comprehending the morphology and other structural characteristics. It could be seen from Figure 3, that the morphological features of the native PKPI samples contained a high degree of heterogeneity due to numerous erratic fragments and disorganized structures linked together. In contrast to the native PKPI, the morphology of PKPI samples exposed to various SC-CO_2_ processing conditions was characterized by reduced particle size, loose and tiny fragments with greater homogeneity in particle size distribution. When the processing conditions were changed to a supercritical state of 60 °C (ST 60) or above, carbon dioxide under the supercritical state induced denaturation and aggregation through covalent and non-covalent interactions due to high electrostatic repulsions between the protein molecules that increased surface charge and resulting in a conglomeration of tiny aggregates. Additionally, a remarkable change in particle size distribution was noticed in the microstructure of SC-CO_2_-treated PKPI samples, demonstrating that carbon dioxide under a supercritical state at higher temperatures induced homogenization. These microstructural adaptations in PKPI samples might also be ascribed to the greater solubility of carbon dioxide in conjunction with the temperature destabilizing the water protein intercommunications and resulting in considerable alterations in protein structure [31]. The findings are in line with Murtaza et al. [24], who suggested that the hydrophilic nature of pressurized carbon dioxide in the supercritical state affects the stability of water–protein bonding, resulting in structural changes in protein molecules. Striolo et al. [46] highlighted that the formation of amide bonds by the interaction of carbon dioxide with amine groups in specific amino acid residues alters the protein’s secondary structure, biological activity, and subsequent morphology. According to Malik and Saini [19], the changes in microstructure might be the consequence of protein molecule expansion brought about by increased exposure of embedded hydrophobic groups and free-SH groups due to the structural unfolding of proteins, which alters the interactions and changes in morphological characteristics. Sheng et al. [40] suggested that high-pressure carbon dioxide treatment induced the unfolding of the partial protein conformation, exposed buried inner sulfhydryl groups, and even disrupted disulfide bonds, which changed the morphology of a liquid whole egg. Hence, the protein molecules underwent dissociation and structural changes during SC-CO_2_ treatments that influenced the microstructure of PKPI samples. The overall results indicated that the protein molecules during SC-CO_2_ treatments dissociated, aggregated, and were dependent on SC-CO_2_ treatment temperature.

### 3.4. X-ray Diffraction (XRD) Pattern of PKPIs

Biopolymers primarily comprise crystalline and amorphous portions, and the ratio of these portions influences the polymer’s structural properties [5]. The effect of SC-CO_2_ treatment on structural characteristics such as crystalline, semi-crystalline, and amorphous structures of the PKPI samples was examined employing the XRD analysis. XRD patterns of the native PKPI samples exhibited a weak peak at a Bragg value of around 11°, and the intensity of this peak decreased with increasing the treatment temperature (Figure 4). Additionally, a sharp and broad peak at a Bragg value of around 20° was visible in the XRD patterns of native and SC-CO_2_-treated PKPI samples, indicating that more non-crystalline materials were present in all PKPIs. Malik and Saini [19] recorded similar peaks for sunflower protein isolates. The strength of this peak also decreased as the temperature was raised, reflecting a decline in the proportion of amorphous substances. The reduction in intensities and shifting of these observed peaks indicates a partial denaturation of SC-CO_2_-treated PKPI samples because these peaks correspond to the α-helical and *β*-sheet structures of the protein molecules [47]. The existence of α-helical and *β*-sheet structures in the polypeptide chain structure is indicated by two small sharp peaks at around 2θ = 8.90° and 2θ = 20.14° in the XRD pattern of the isolated proteins [48]. Compared to native PKPI, crystalline percentages in SC-CO_2_-treated PKPIs slightly declined. A minimum percentage of 30.5 was noticed in ST 70-treated samples, which was 1.21 times lower than the native PKPI sample (Figure 4). With the increase in treatment temperature, the crystallite size decreased, and PKPIs treated at 70 °C (ST 70) demonstrated a decrease of 20.74% in crystallite size. The loss of crystal arrangement was confirmed by the decreased X-ray peak intensity, which was most likely caused by the disrupted hydrogen bond, resulting in the displacement of nearby double helices and imperfect parallel orientation rearrangement [49]. A 10.81% loss of crystallinity was observed in PKPIs treated at 70 °C (ST 70), which might be ascribed to the conversion of α-helical structures into *β*-sheet structures due to the pressurized carbon dioxide in the supercritical state that affects structural arrangements and subsequently reduces the crystallinity. Additionally, particle size and diffraction intensity are directly correlated, with lesser diffraction intensity indicating smaller protein crystal size and vice versa. The size of the crystal influences the diffraction angle (2θ) and intensity and aid in elucidating probable conformational alterations in protein structure and the interactions of proteins with other molecules [50]. On the contrary, the enhanced amorphous percentage with the increasing treatment temperature indicates better solubility and water-holding capacity of the SC-CO_2_-treated PKPIs [47]. The findings are in line with those of Malik and Saini [19]; Mir et al. [5], and Li et al. [47] for sunflower protein isolates, quinoa protein isolates, and canola seed proteins, respectively. Therefore, XRD findings allow us to predict the pattern of structural changes in PKPI during SC-CO_2_ treatment.

### 3.5. Functional Characterization of PKPI

#### 3.5.1. Effect of SC-CO_2_ Treatment on Dispersibility

Dispersibility typically refers to the reconstitution potential of a protein, which is primarily influenced by the solvents’ ionic content, pH, temperature and agitation level [51]. Higher reconstituteability of proteins improves other techno-functional properties, including emulsifying and foaming capacity [52]. PKPI samples treated with SC-CO_2_ showed significantly (*p* < 0.05) higher values of dispersibility than native PKPI samples (Table 1). The dispersibility values increased with increasing processing temperature, and the highest value was recorded at 60 °C (ST 60), which was 1.15-fold higher than the native PKPI sample. Since solubility and dispersibility are closely related, the higher dispersibility in SC-CO_2_-treated PKPI samples at 60 °C indicates excellent solubility of SC-CO_2_-treated PKPIs, which is an excellent index for their functionality because high solubility allows proteins to have a much wider range of potential applications. The overall improvement in dispersibility might be ascribed to the dissociation of the large insoluble protein aggregates into soluble supramolecular aggregates due to the greater solubility of carbon dioxide that causes considerable changes in protein conformation and enhances the reconstitutability of PKPI treated with SC-CO_2_. Dispersibility is connected to particle size, and structural modification is considered to influence protein rehydration characteristics [53]. The results are supported by the outcome of Malik and Saini [52], and Mir et al. [35] for sunflower seed protein isolates and *Chenopodium* seed protein isolates, respectively, and reported that the dispersibility is directly related to the solubility and higher protein dispersibility enhances other techno-functional properties including the emulsifying and foaming abilities of proteins.

#### 3.5.2. Effect of SC-CO_2_ Treatment on Least Gelation Concentration (LGC)

Gelation is the aggregation of the denatured proteins into a three-dimensional network with adsorbed water and the ability to produce a stiff gel structure depending on the critical concentration, protein–protein interaction, and interaction with other non-protein components [54]. LGC is indicative of the gelation capacity of proteins; the lower the LGC, the better the gelling ability of proteins. In the current study, for native PKPI samples, no gel was formed at a concentration ranging from 6% to 14%. Nevertheless, a gel was formed at 16% (LGC) (Table 1). Compared to the native sample, SC-CO_2_-treated PKPIs at processing temperatures of 60 °C or above started to form a gel at a concentration of 12%, indicating a high degree of asymmetry in protein structure and a better gelling ability. The variation in LGC values of SC-CO_2_-treated PKPI samples could be attributed to the partial unfolding of proteins because the carbon dioxide under a supercritical state at higher temperatures has a more robust diffusion capacity that influences the protein–protein and protein–water interactions due to changes in structural characteristics of protein molecules that stabilize the gels. Protein functionality is often associated with changes in secondary and tertiary structure and SC-CO_2_ treatment have altered the structure that initiated the formation of a randomly aggregated protein network by physical interactions and subsequently improved the strength of gel. The observations for the gelation capacity of PKPI samples align with Peyreno et al. [55] for cowpea protein isolates during thermal and high hydrostatic pressure treatment. The observed changes in LGC values could also be attributed to differences in the thermal stability of PKPI samples because carbon dioxide under a supercritical state reacts with the amide groups of some amino acid residues and forms amino bonds, resulting in considerable changes in protein conformation and thermal stability that influences the reconstitution potential of PKPI samples [24]. The change in thermal stability of PKPI samples is also supported by the results of thermo-gravimetric analysis (Figure 2). Chandra et al. [56] reported that the gelation capacity of proteins also depends upon the hydrophobicity and sulfhydryl groups of the protein and protein gelation in food items is critical for achieving desired sensory and textural characteristics. Carbon dioxide under a supercritical state induces changes in the protein structure by lowering the intermolecular associations and exposing buried sulfhydryl groups that change the gelation capacity of SC-CO_2_-treated PKPI samples [33].

### 3.6. Effect of SC-CO_2_ Treatment on Water Activity

It is well recognized that water activity (a_W_) is an essential characteristic for determining the stability of foods. Knowing the optimal moisture levels and the hydrodynamics of the water–protein behavior would be advantageous in determining the shelf-life of protein isolates [57]. Compared to the native sample, the SC-CO_2_-treated PKPI samples showed a significant (*p* < 0.05) increase in a_W_ (Table 1). The extension of processing temperature caused remarkable changes in a_W_, and PKPI samples processed at 50 °C displayed the highest a_W_. The increase in a_W_ might be attributed to an unfolding of protein structure that exposed additional sorption sites, causing water molecules to bind to the surface area of PKPI. Water sorption is often associated with protein surface area, and numerous studies have found that increasing the surface area tends to improve water adsorption by unfolding the globular protein and releasing more charged polar groups and carboxyl groups [57]. When the processing conditions were shifted to a supercritical state of 60 °C (ST 60) or higher, the a_W_ of PKPI decreased, indicating a paucity of polar amino acids on the surface of protein molecules, which are crucial for protein–water interactions. The decrease in water activity at higher processing temperatures could be ascribed to the formation of soluble aggregates that progressively disrupts ionic bonding, and result in the reduction of protein–water interactions. The findings are consistent with Al-Jassar et al. [57], who suggested that the structural alteration of protein powder alters the protein–water interactions and exposes more hydrophilic groups after heat treatment that increases a_W_. The observed differences in a_W_ of SC-CO_2_-treated PKPIs could be ascribed to the dissolved carbon dioxide that interacted with PKPIs during the processing and structural changes [31].

## 4. Conclusions

The recovery of plum kernel proteins is of critical relevance in human nutrition. To increase its functionality in commercial applications, plum kernel protein isolate was treated with SC-CO_2_ treatment at various processing conditions. The results suggest that SC-CO_2_ treatment modified the techno-functional characteristics of plum kernel protein isolates owing to the disruption of the protein–protein interactions, destabilization of entanglement network, and enhancement of protein molecule slippage. The increase in processing temperature caused a remarkable improvement in the reconstitution potential and the gelation temperature of all SC-CO_2_-treated PKPI samples. PKPIs treated at 60 °C (ST 60) showed the highest dispersibility, which was 1.15-fold higher than the native PKPI sample. Moreover, all the SC-CO_2_-treated PKPIs were more thermally stable than the native PKPI, indicating an increased heat requirement for thermal denaturation. The loss of crystallinity with the increasing temperature indicates the structural arrangements in SC-CO_2_-treated PKPIs owing to the rapid wetting and penetration of supercritical carbon dioxide into the protein molecule, which interferes with its local conformation. The morphology analysis revealed loose and tiny fragments in SC-CO_2_-treated PKPIs with greater homogenization. Therefore, using appropriate SC-CO_2_ settings, i.e., system temperature of 60 °C, a pressure of 20 MPa, a CO_2_ flow rate of 20 g/min, and a processing period of 60 min would be a substantial and environmentally acceptable approach to improve the techno-functional properties of proteins and extend their use in the food processing industry.

## Figures and Tables

**Figure 1 foods-12-00815-f001:**
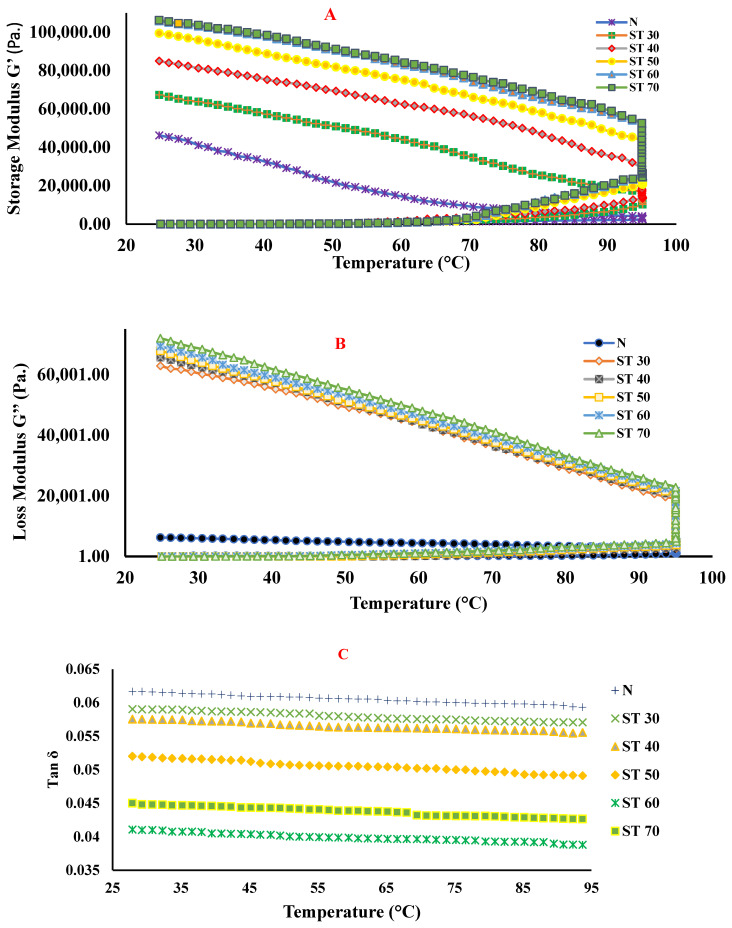
**(i).** Effect of supercritical carbon dioxide (SC-CO_2_) treatment on gelation dynamics of plum kernel protein isolates (**A**): storage modulus, (**B**): loss modulus, (**C**): loss tangent as a temperature, N: native plum kernel protein isolate; ST 30: SC-CO_2_-treated plum kernel protein isolate at 30 °C; ST 40: SC-CO_2_-treated plum kernel protein isolate at 40 °C; ST 50: SC-CO_2_-treated plum kernel protein isolate at 50 °C; ST 60: SC-CO_2_-treated plum kernel protein isolate at 60 °C and ST 70: SC-CO_2_-treated plum kernel protein isolate at 70 °C. **(ii).** Effect of supercritical carbon dioxide (SC-CO_2_) treatment on gelation dynamics of plum kernel protein isolates (**D**): frequency sweep teat as a function of storage modulus, and (**E**): frequency sweep teat as a function of loss modulus. N: native plum kernel protein isolate; ST 30: SC-CO_2_-treated plum kernel protein isolate at 30 °C; ST 40: SC-CO_2_-treated plum kernel protein isolate at 40 °C; ST 50: SC-CO_2_-treated plum kernel protein isolate at 50 °C; ST 60: SC-CO_2_-treated plum kernel protein isolate at 60 °C and ST 70: SC-CO_2_-treated plum kernel protein isolate at 70 °C.

**Figure 2 foods-12-00815-f002:**
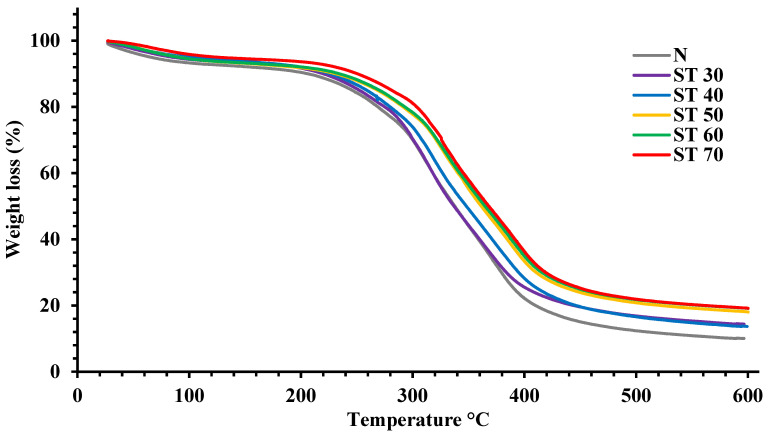
Thermal gravimetric analysis of N: native plum kernel protein isolate; ST 30: supercritical carbon dioxide (SC-CO_2_)-treated plum kernel protein isolate at 30 °C; ST 40: SC-CO_2_-treated plum kernel protein isolate at 40 °C; ST 50: SC-CO_2_-treated plum kernel protein isolate at 50 °C; ST 60: SC-CO_2_-treated plum kernel protein isolate at 60 °C and ST 70: SC-CO_2_-treated plum kernel protein isolate at 70 °C.

**Figure 3 foods-12-00815-f003:**
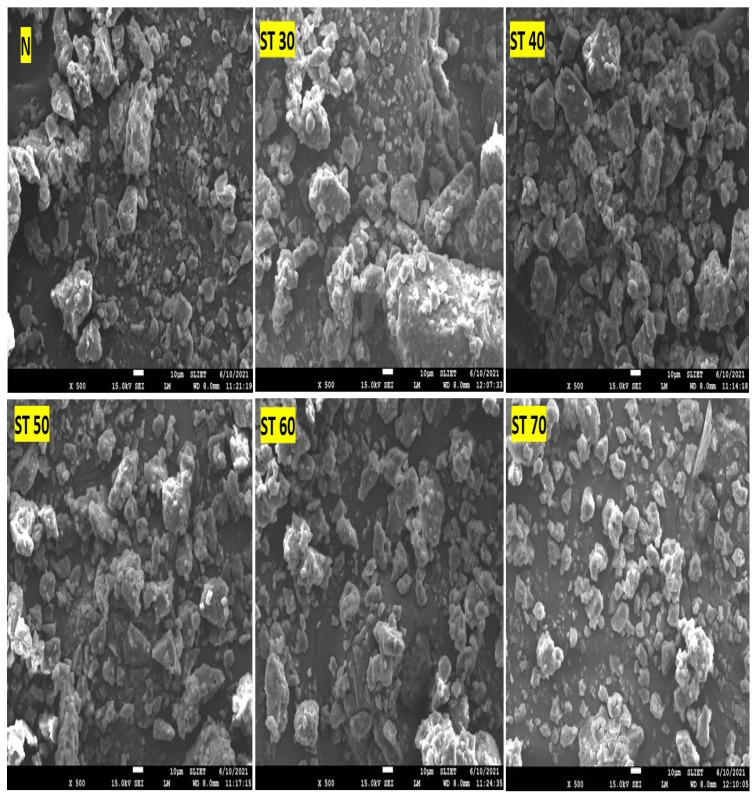
SEM micrographs of N: native plum kernel protein isolate; ST 30: supercritical carbon dioxide (SC-CO_2_)-treated plum kernel protein isolate at 30 °C; ST 40: SC-CO_2_-treated plum kernel protein isolate at 40 °C; ST 50: SC-CO_2_-treated plum kernel protein isolate at 50 °C; ST 60: SC-CO_2_-treated plum kernel protein isolate at 60 °C and ST 70: SC-CO_2_-treated plum kernel protein isolate at 70 °C.

**Figure 4 foods-12-00815-f004:**
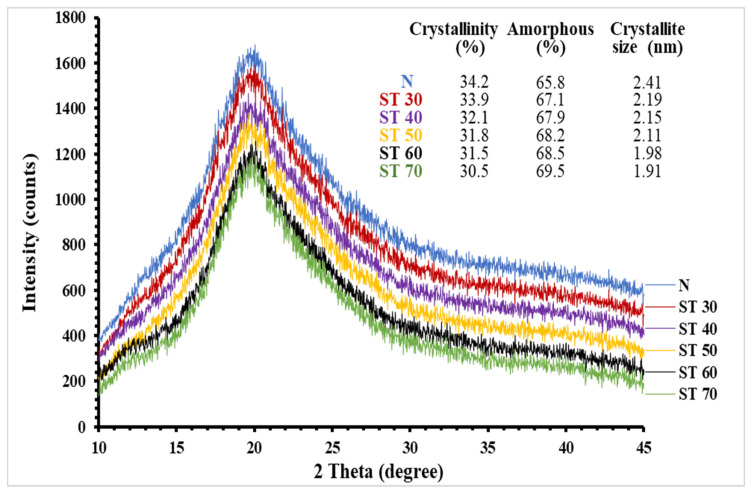
XRD of N: native plum kernel protein isolate; ST 30: supercritical carbon dioxide (SC-CO_2_)-treated plum kernel protein isolate at 30 °C; ST 40: SC-CO_2_-treated plum kernel protein isolate at 40 °C; ST 50: SC-CO_2_-treated plum kernel protein isolate at 50 °C; ST 60: SC-CO_2_-treated plum kernel protein isolate at 60 °C and ST 70: SC-CO_2_-treated plum kernel protein isolate at 70 °C.

**Table 1 foods-12-00815-t001:** Effect of SC-CO_2_ treatment on dispersibility, least gelation concentration, and water activity (a_w_) of PKPIs.

Treatments	Dispersibility (%)	Least Gelation Concentration (%)	Water Activity (a_W_)
N	62.09 ± 0.18 ^f^	16 ± 0.00 ^a^	0.36 ± 0.01
ST 30	63.78 ± 0.23 ^e^	16 ± 0.00 ^a^	0.38 ± 0.02 ^e^
ST 40	65.79 ± 0.21 ^d^	14 ± 0.00 ^b^	0.41 ± 0.01 ^b^
ST 50	69.30 ± 0.24 ^c^	14 ± 0.00 ^b^	0.43 ± 0.01 ^a^
ST 60	71.99 ± 0.12 ^a^	12 ± 0.00 ^c^	0.42 ± 0.01 ^c^
ST 70	70.15 ± 0.21 ^b^	12 ± 0.00 ^c^	0.40 ± 0.02 ^d^

Means ± standard deviation values of triplicates. ^a–f^ column values followed by the same superscript letter are not significantly different (*p* < 0.05). N: native plum kernel protein isolate; ST 30: supercritical carbon dioxide (SC-CO_2_)-treated plum kernel protein isolate at 30 °C; ST 40: SC-CO_2_-treated plum kernel protein isolate at 40 °C; ST 50: SC-CO_2_-treated plum kernel protein isolate at 50 °C; ST 60: SC-CO_2_-treated plum kernel protein isolate at 60 °C and ST 70: SC-CO_2_-treated plum kernel protein isolate at 70 °C.

## Data Availability

The data that support the findings of this study are available from the corresponding authors upon reasonable request.

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
