# Peer review of "Investigating the Effect of Supercritical Carbon Dioxide Treatment on the Rheological, Thermal, and Functional Properties of Plum (Prunus domestica L.) Kernel Protein Isolates"

_foods, 2023, doi:10.3390/foods12040815_

Round 1

Reviewer 1 Report

Dear Author

The paper presented by the author is evaluated as an effort to find a food source or functional material that is lacking in the future society, and I think it presents a feasible goal for the present. Also, I am delighted as a reviewer who evaluates papers presenting such results. However, I think that the following questions need to be supplemented.

1. In the SC-CO2 treatment, a basic explanation of the selection of temperature treatment at 30-70℃ should be included in the introduction or experimental method.

2. In Figure 1C, you want to display the legend in the same arrangement as other A, B, D, E.

3. I think that the expression that heat resistance decreases with temperature in TGA characteristics is correct in terms of data. However, if it is used as food, it is not possible to use it above 400℃. Doesn't it need more explanation at the temperature that is actually usable? And there is no significance comparison with one comparison. Is there no need to supplement this?

4. It is difficult to see objective differences in Figure 3. I wish I could explain with more concrete results.

5. There is no indication of ST-70 in the respective XRD chromatograms in Figure 4, and there are no values for ST-50, ST-60, ST-70 in the Crystalllite size indication. Please fill in the values and explain.

6. The result of Figure 4 is mainly data tracking the change in crystallinity, but I think that the difference in crystallinity is actually minute. Do you think there are as many differences as the authors describe?

7. In Table 1, ST-60 has higher dispersibility than all samples, but there is no difference between ST-70 and minimum gelation concentration. On the other hand, ST-50 is rather higher in water activity. Considering these, I think a full explanation is necessary.

Author Response

The authors are grateful to the Reviewer for their comprehensive and insightful assessment of the manuscript. The responses to the comments are attached, and the changes are highlighted in the manuscript.

Reviewer 2 Report

This manuscript is interesting. However, minor suggestions are needed to improve the manuscript as follows:

- In Introduction part: More critical comments on the effects of SC-CO2 treatment conditions on the rheological, thermal, and functional properties?.

- Please check and revise all Figure 4 and Table 1 regarding legends, and explanations, etc.

Author Response

The authors are grateful to the Reviewers for their comprehensive and insightful assessment of the manuscript. The responses to the comments are attached, and the changes are highlighted in the manuscript.

Reviewer 3 Report

This manuscript dealt with investigating the effect of supercritical carbon dioxide treatment on the rheological, thermal, and functional properties of plum (Prunus domestica L.) kernel protein isolates. This paper will be of interest to the readers of the journal. Some questions need to be addressed.

Q: What is the purpose of the supercritical carbon dioxide treatment on PKPI beside modifying the rheological, thermal, and functional properties? 

Q: Figure 1 A and B: these figures look weird to me. I don’t think it is possible to obtain different G’ or G” values at the same measurement temperature? These figures are not convincing.

Q: What are the food applications of the PKPI? 

Author Response

The authors are grateful to the Reviewer for their comprehensive and insightful assessment of the manuscript. The responses to the comments are attached below, and the changes are highlighted in the manuscript.

Reviewer 4 Report

Dear Authors,
a great topic related to the use of post-production waste. Especially since you mention the possibility of using research in industry "...extend its use in food and non-food applications...". There is only one detail that is relatively important. What is the protein concentration in plum kernel protein isolate. The answer to this question may be crucial for the obtained results. The question is what was analyzed almost pure protein or isolate protein preparation. If the isolate, what was the concentration of protein, what was the content of non-protein substances. On this basis, it is possible to infer the characteristics of the preparation (not the proteins). Where are the interactions? You provide information about gelling. What gelled? How much was it? Did it react with other ingredients? Which ones? Etc, etc.

Author Response

(The authors gave the same response as above.)

Round 2

Reviewer 3 Report

The authors have not properly addressed my concerns. Fig. 1A and B are not convincing.

Author Response

Response to Reviewer 3 Comments

Point 1. What is the purpose of the supercritical carbon dioxide treatment on PKPI besides modifying the rheological, thermal, and functional properties? 

Response 1: Supercritical carbon dioxide (SC-CO2) exhibits a plethora of desirable characteristics for application in the food industry. The SC-CO2 technique has been recognized as a novel technique that has raised interest in its application owing to its valuable characteristics, including non-toxicity, colourless, odourless, low-cost, non-flammable, non-explosive, non-polluting, recoverable characteristics, green, liquid-like density, gas-like diffusivity and low consumption of chemical compounds. The SC-CO2 is a chemical-free processing method that has become an excellent technique for extracting, refining, and modifying biomolecules at the industrial level. SC-CO2 technique has been employed for the fractionation of whey protein isolate to produce enriched α-lactalbumin and β-lactoglobulin food ingredients, and the main advantage is that depressurization of the system releases the dissolved gas and returns the pH of the products to a value close to the initial pH, without the addition of any contaminants. Moreover, the SC-CO2 technique offers many advantages over conventional thermal processing methods, including the ability to retain the quality of food, which may be lost during conventional thermal processing.

Point 2. Figure 1 A and B: these figures look weird to me. I don't think it is possible to obtain different G' or G" values at the same measurement temperature? These figures are not convincing.

Response 2: The authors regret that figures 1A and B of the manuscript were somehow inadequate. The change in storage modulus (G') and loss modulus (G") of plum kernel protein isolate at the same temperature could be ascribed to the increased aggregation of proteins, indicating stronger and more elastic gels. The results of G', and G" of plum kernel protein isolate are consistent with (Lamsal et al., 2007; Malik & Saini, 2017) who reported that all sunflower seed protein isolates gels were having higher G' values than G" throughout the test, suggesting a predominant elastic behaviour. Similarly, (Malik & Saini, 2019; Mir et al., 2020) reported a higher magnitude storage modulus (G') than loss modulus (G")  for all the protein isolates of Chenopodium album protein isolates.

Lamsal, B. P., Jung, S., & Johnson, L. A. (2007). Rheological properties of soy protein hydrolysates obtained from limited enzymatic hydrolysis. LWT, 40(7), 1215–1223. https://doi.org/10.1016/j.lwt.2006.08.021

Malik, M. A., & Saini, C. S. (2017). Polyphenol removal from sunflower seed and kernel: Effect on functional and rheological properties of protein isolates. Food Hydrocolloids, 63, 705–715. https://doi.org/10.1016/j.foodhyd.2016.10.026

Malik, M. A., & Saini, C. S. (2019). Heat treatment of sunflower protein isolates near isoelectric point: Effect on rheological and structural properties. Food Chemistry, 276, 554–561. https://doi.org/10.1016/j.foodchem.2018.10.060

Mir, N. A., Riar, C. S., & Singh, S. (2020). Structural modification in album (Chenopodium album) protein isolates due to controlled thermal modification and its relationship with protein digestibility and functionality. Food Hydrocolloids, 103, 105708. https://doi.org/10.1016/j.foodhyd.2020.105708

Point 3.  What are the food applications of the PKPI? 

Response 3: Plums kernels are cheap sources of proteins that are irretrievably lost after canning and beverage production. The recovery of these plum proteins from agro-food wastes could be profitably adopted in human nutrition. Plum kernel protein isolate (PKPI) has shown desirable functional properties (solubility, dispersity, water and oil adsorption capacity, foaming and emulsifying ability) that can be added directly to foods to improve their functionality (Sheikh et al., 2023) like the ability to form or stabilize emulsions, the ability to create or stabilize foams, solubility, gelation,  water binding, and fat binding properties. The protein isolate obtained from plum kernel cake is a potential source of natural products for food applications, like a functional food additive with good digestibility and beneficial bioactive peptides that positively affect human health (Cakarevic et al., 2019). A potential application of PKPI could be the production of bioactive peptides. A bioactive peptide is a functional ingredient if it has successfully demonstrated its beneficial effect on one or more functions of the body beyond its nutritional effects so that its effect is significant in reducing the risk of suffering a disease (González-García et al., 2016). The use of bio-based films and coatings from proteins as an alternative to non-biodegradable polymeric films often exhibits a diverse range of applications in the food sectors because of their superior biodegradability, biocompatibility, edibility, organoleptic features, and provide an effective barrier to the mass transfer (e.g., moisture, gases, or flavour or aromas) between the phases of composite products as well as between the product and surrounding medium.

ÄŒakarević, J., Vidović, S., Vladić, J., Jokić, S., Pavlović, N., & Popović, L. (2019). Plum oil cake protein isolate: A potential source of bioactive peptides. Food and Feed Research, 46(2), 171–178. https://doi.org/10.5937/ffr1902171c

González-García, E., Marina, M. L., García, M. C., Righetti, P. G., & Fasoli, E. (2016). Identification of plum and peach seed proteins by nLC-MS/MS via combinatorial peptide ligand libraries. Journal of Proteomics, 148, 105–112. https://doi.org/10.1016/j.jprot.2016.07.024

Sheikh, M. A., Singh, C., & Kumar, H. (2023). Structural modification of plum ( Prunus domestica L ) kernel protein isolate by supercritical carbon-dioxide treatment : Functional properties and in-vitro protein digestibility. International Journal of Biological Macromolecules, 230(January), 123128. https://doi.org/10.1016/j.ijbiomac.2022.123128

Reviewer 4 Report

Dear Authors,

thank you for your replies. Please, place the information contained in the answers, especially in response 1 regarding the protein concentration in the text of the publication.

Author Response

Response to Reviewer 4 Comments

Point 1.  What is the protein concentration in plum kernel protein isolate. The answer to this question may be crucial for the obtained results. The question is what was analyzed almost pure protein or isolate protein preparation. If the isolate, what was the concentration of protein, what was the content of non-protein substances. On this basis, it is possible to infer the characteristics of the preparation (not the proteins). Where are the interactions?

Response 1: The authors appreciate the reviewers' comprehensive and insightful assessment of the manuscript and are grateful for the constructive feedback. The flow sheet for the isolation of proteins from plum kernels and their characterization is demonstrated in our previously published article (Sheikh et al., 2023).

Sheikh, M. A., Singh, C., & Kumar, H. (2023). Structural modification of plum ( Prunus domestica L ) kernel protein isolate by supercritical carbon-dioxide treatment : Functional properties and in-vitro protein digestibility. International Journal of Biological Macromolecules, 230, 123128. https://doi.org/10.1016/j.ijbiomac.2022.123128.

The compositional characteristic of protein isolates (extraction at pH 10.2) from plum kernels includes protein content = 88.09 ± 0.45 %, moisture content = 8.16 ±0.11 %, ash content = 1.05 ± 0.03 %, crude fibre = 1.18 ± 0.08 %,  and carbohydrates = 0.68 ± 0.02 %. Fat content in the protein isolates was not detected because defatted plum kernel meal was used for protein extraction. The purity and yield of plum kernel protein isolate were 88.09 % and 24.6 %  at a pH of 10.2, respectively. A significant increase in the protein yield was observed at higher pH of 10.2, which suggests that the solubility of the proteins increased in more extreme alkaline conditions. In the presence of alkaline conditions, the proteins become increasingly negatively charged due to the ionization of the carboxylic groups and deprotonation of the amine group, which enhances the electrostatic repulsion between negatively charged proteins. This increases protein-water interactions and, thereby, protein solubility. Further increase in alkaline conditions decreased the protein purity from 88.09 to 85.24%, which could be ascribed to the increase in the amount of non-protein components co-precipitating with the plum kernel protein isolates.

Point 2.  You provide information about gelling. What gelled? How much was it? Did it react with other ingredients? Which ones? Etc, etc.

Response 2: The ability of proteins to form a gel is based on the amphiphilic nature of proteins and is an important functionality to provide a structural matrix for holding water, flavors, sugars, and other ingredients in many food applications. Protein gels are composed of three-dimensional networks of inter-twined, partially associated polypeptides, and are characterized by a relative high viscosity, plasticity, and elasticity. The gelation of protein dispersions can be caused by heating and cooling, followed by a critical balance of attractive and repulsive forces is attained. Heat treatment is a common gelling strategy and protein cold-set gelation is performed by heating the proteins in the solution, followed by a cooling step. Similarly, a protein dispersion of 20% (w/v) was heated at a rate of 10 °C/min from 25 °C to 95 °C, kept there for 15 minutes, and then cooled down slightly to 25°C at the same rate. All protein dispersions exhibited gelation characteristics. However, the results of supercritical carbon-dioxide treated plum kernel protein isolate (PKPI) exhibited better gelation property than the native PKPI samples. The percentage of other non-protein substances were negligible, so the interaction with other ingredients was not observed. The thermal gelation of proteins is a complex process involving (i) heat-induced unfolding of the native protein structure exposing the interaction sites such as sulfhydryl, hydrophobic, hydrogen-bonding, and electrostatic groups (ii) aggregations of unfolded proteins through hydrophobic interactions, and formation of disulphide bonds, and (iii) agglomeration of aggregates forming a three-dimensional network.
